# Prevalence of Antibodies to SARS-CoV-2 in Italian Adults and Associated Risk Factors

**DOI:** 10.3390/jcm9092780

**Published:** 2020-08-27

**Authors:** Antonio Vena, Marco Berruti, Andrea Adessi, Pietro Blumetti, Michele Brignole, Renato Colognato, Germano Gaggioli, Daniele Roberto Giacobbe, Luisa Bracci-Laudiero, Laura Magnasco, Alessio Signori, Lucia Taramasso, Marco Varelli, Nicoletta Vendola, Lorenzo Ball, Chiara Robba, Denise Battaglini, Iole Brunetti, Paolo Pelosi, Matteo Bassetti

**Affiliations:** 1Infectious Diseases Unit, San Martino Policlinico Hospital—IRCCS for Oncology and Neurosciences, 16132 Genoa, Italy; anton.vena@gmail.com (A.V.); marco.berruti1@gmail.com (M.B.); daniele.roberto.giacobbe@gmail.com (D.R.G.); lmagnasco90@gmail.com (L.M.); taramasso.lucia@gmail.com (L.T.); 2Department of Health Sciences (DISSAL), University of Genoa, 16132 Genoa, Italy; 3Onilab Milano, 20123 Milano, Italy; adessi@onilab.it; 4Medical Center srl, Sesto Calende, 21018 Varese, Italy; Pietro.blumetti@medicalcentersrl.it (P.B.); renato.colognato@gmail.com (R.C.); 5Department of Cardiology, Arrhytmology Centre and Syncope Unit, Ospedale del Tigullio, 16033 Lavagna, Italy; mbrignole@outlook.it; 6IRCCS Istituto Auxologico Italiano, Faint and Fall Programme, Ospedale San Luca, 20149 Milano, Italy; 7Division of Cardiology, Ospedale Villa Scassi, 16149 Genova, Italy; gaggioligermano@gmail.com; 8Institute of Translational Pharmacology, Consiglio Nazionale Delle Ricerche (CNR), 00185 Rome, Italy; luisa.braccilaudiero@ift.cnr.it; 9Division of Rheumatology and Immuno-Rheumatology Research Laboratories, Bambino Gesù Children’s Hospital, 00165 Rome, Italy; 10Section of Biostatistics, Department of Health Sciences, University of Genova, 16132 Genova, Italy; alessio.signori.unige@gmail.com; 11Diagnostic Institute Varelli, Clinical Analysis, 80126 Napoli, Italy; marco.varelli@istitutovarelli.it; 12Division of Obstetrics and Gynecology, Sant’Andrea Hospital, 13100 Vercelli, Italy; vendola.n@yahoo.com; 13Department of Surgical Sciences and Integrated Diagnostics, University of Genoa, 16132 Genoa, Italy; lorenzo.loryball@gmail.com (L.B.); ppelosi@hotmail.com (P.P.); 14Anesthesia and Intensive Care, San Martino Policlinico Hospital, IRCCS for Oncology and Neurosciences, 16132 Genoa, Italy; kiarobba@gmail.com (C.R.); battaglini.denise@gmail.com (D.B.); brunettimed@gmail.com (I.B.)

**Keywords:** SARS-CoV-2, COVID-19, antibodies, serological test

## Abstract

We aimed to assess the prevalence of and factors associated with anti- severe acute respiratory syndrome coronavirus-2 (SARS-CoV-2) positivity in a large population of adult volunteers from five administrative departments of the Liguria and Lombardia regions. A total of 3609 individuals were included in this analysis. Participants were tested for anti-SARS-CoV-2 antibodies [Immunoglobulin G (IgG) and M (IgM) class antibodies] at three private laboratories (Istituto Diganostico Varelli, Medical Center, and Casa della Salute di Genova). Demographic data, occupational or private exposure to SARS-CoV-2-infected patients, and prior medical history consistent with SARS-CoV-2 infection were collected according to a preplanned analysis. The overall seroprevalence of anti-SARS-CoV-2 antibodies (IgG and/or IgM) was 11.0% [398/3609; confidence interval (CI) 10.0%–12.1%]. Seroprevalence was higher in female inmates than in male inmates (12.5% vs. 9.2%, respectively, *p* = 0.002), with the highest rate observed among adults aged >55 years (13.2%). A generalized estimating equations model showed that the main risk factors associated with SARS-CoV-2 seroprevalence were the following: an occupational exposure to the virus [Odd ratio (OR) = 2.36; 95% CI 1.59–3.50, *p* = 0.001], being a long-term care facility resident (OR = 4.53; 95% CI 3.19–6.45, *p* = 0.001), and reporting previous symptoms of influenza-like illness (OR = 4.86; 95% CI 3.75–6.30, *p* = 0.001) or loss of sense of smell or taste (OR = 41.00; 95% CI 18.94–88.71, *p* = 0.001). In conclusion, we found a high prevalence (11.0%) of SARS-CoV-2 infection that is significantly associated with residing in long-term care facilities or occupational exposure to the virus. These findings warrant further investigation into SARS-CoV-2 antibody prevalence among the Italian population.

## 1. Introduction

In Italy, the first case of pandemic severe acute respiratory syndrome coronavirus-2 (SARS-CoV-2) infection was reported on 20 February, 2020. Since then, the number of cases increased rapidly in the north of the country, with the Lombardia and Liguria regions being heavily affected by the infection [1]. By the end of April 2020, approximately 85,000 laboratory confirmed cases -of SARS-CoV-2 infection were reported in this geographical area of the country [2]. However, these data included only a fraction of the real number of SARS-CoV-2 infections, since not all infected patients were symptomatic [3,4,5], required hospitalizations, or provided specimens for laboratory testing. The extent to which surveillance data reflect the true burden of the disease can also be affected by changes in laboratory testing recommendation [1]. Serology can represent a key element to overcoming these limits and to better understanding the infection statistics at a population level. The primary outcome of this study was to estimate the prevalence of SARS-CoV-2 antibodies. The secondary outcome was to evaluate possible factors associated with anti-SARS-CoV-2 positivity in a large population of individuals from five administrative departments of the Liguria and Lombardia regions.

## 2. Experimental Methods

This was an observational study designed to evaluate the prevalence and factors associated with SARS-CoV-2 infections among voluntary, unpaid individuals tested for SARS-CoV-2 antibodies in three private institutions (Istituto Diagnostico Varelli, Medical Center, and Casa della Salute di Genova) during March and April 2020. These institutions altogether include approximately 5,784,974 inhabitants living in five administrative departments (Milano, Varese, Pavia, Genova and Savona) of the Liguria and Lombardia regions. Each laboratory process, about 500,000 samples per year, offers a comprehensive range of tests including clinical biochemistry, serology, and genetic analysis.

### 2.1. Participants

We included non-hospitalized participants (aged > 18 years) who voluntarily tested for SARS-CoV-2 antibodies in an outpatient setting. After providing informed consent, a sample of venous blood was collected from each participant, all of whom also completed a questionnaire on potential risk factors for developing SARS-CoV-2 infection. Recorded data included age, sex. and occupational or private exposure to SARS-CoV-2 infected patients. In addition, information regarding stays at a long-term care facilities or prior medical history consistent with SARS-CoV-2 infection (influenza-like illness defined according to WHO criteria [6] or loss of smell or taste) within the previous month, were also collected.

### 2.2. Endpoint

The primary goal was to assess the prevalence of SARS-CoV-2 antibodies [either Immunoglobulin M (IgM) and G (IgG)] positivity among the study population. The secondary goal was to investigate the association between positive tests and demographics (age and sex), occupational and private contact with SARS-CoV-2 infected patients, living in long-term care facilities, and prior symptoms consistent with SARS-CoV-2 infection.

### 2.3. Detection of Infection

Blood samples were analyzed for serological detection at each participating laboratory by trained staff, unaware of the clinical details of the tested patients. The first laboratory (Istituto Diagnostico Varelli) used a chemiluminescent quantitative immunoassay detecting antibodies against nucleocapsid protein and spike protein (the MaglumiTM 2019) [7]. According to the manufacturer’s recommendations, samples were considered positive above a threshold of 1.1 AU/mL for IgM and IgG. This cut-off resulted in clinical sensitivities/specificities of 78.6%/97.5% and 91.2%/97.3% for IgM and IgG, respectively [7,8]. The second laboratory (Medical Center) applied a rapid chromatographic immunoassay for the qualitative detection of IgG and IgM antibody against spike protein (Realy tech^®^ 2019 nCOV/COVID-19 IgG/IgM Rapid Test Device). The manufacturer’s reported a clinical sensitivity of 92% for IgM; 96% for IgG; and a specificity of 100% for IgM and IgG. The third laboratory (Casa della salute di Genova) assessed anti-SARS-CoV-2 antibodies using a commercially available point-of-care lateral flow immunoassay (Biosynex^®^ Covid-19 BSS, Fribourg, Switzerland) that can simultaneously detect IgM and IgG in human blood, with an overall sensitivity of 88.7% and specificity of 90.6% [9]. This qualitative test detected antibodies against nucleocapsid and spike proteins. All laboratories used internal procedures to validate the diagnostic performance of serological tests. In all cases, the results showed values of sensitivity and specificity consistent with those reported by each manufacturer.

### 2.4. Statistical Analysis

All statistics were analyzed using SPSS software. Prevalence of anti-SARS-CoV-2 antibodies (IgM or IgG) was calculated and the exact binomial distribution was used to calculate 95% confidence intervals (CIs). The association between positive SARS-CoV-2 antibodies and study variables was estimated in two steps. First, a general linear univariate analysis was performed using a Chi-squared test. The second step used a generalized estimating equation (GEE) model to consider laboratory provenience, with SARS-CoV-2 seropositivity used as a dependent variable. Only differences with a *p*-values < 0.05 were considered statistically significant.

### 2.5. Ethical Consideration

The study protocol was approved by the Ethics Committee of Liguria Region (PI Prof. Matteo Bassetti-N. CER Liguria 381/2020-id 10770).

## 3. Results

### 3.1. Participant Demographics and Exposures

Between 1 March and 30 April 2020, 3609 individuals agreed to participate in the study. The mean number of screened individuals per administrative department was 721 (52–1430), representing 12 people per 100,000 inhabitants. The patients’ demographics are outlined in Table 1.

Overall, 55.6% (2007/3609) were women and 44.4% were men (1602/3609). The median age was 51 years [interquartile range (IQR) 41–63], with the age group >55 years being most represented (41.4%; *n* = 1493/3609) and the 18–34 years group being the least represented (15.4%; *n* = 556/3609). All patients lived in the Lombardia and Liguria regions in the administrative departments of Varese (39.6%; *n* = 1430/3609), Pavia (24.1%; *n* = 871/3,609), Milano (21.2%; *n* = 764/3609), Genova (13.6%; *n* = 492/3609,) and Savona (1.4%; *n* = 52/3609;). Approximately 5.7% of the individuals (*n* = 207/3609) lived in a long-term care facility, whereas 5.0% (*n* = 179/3609) reported an occupational exposure to infected patients. When asked about recent medical history, 11.8% (*n* = 427/3609) reported symptoms consistent with influenza-like illness and 0.97% (*n* = 35/3609) reported loss of smell or taste within the previous month.

### 3.2. Prevalence of Sars-CoV-2 Antibodies

Of the 3609 individuals included in the study population, 398 tested anti-SARS-CoV-2 positive [11.0% (CI 10.0%–12.1%)]. Seroprevalence was higher among women vs. men (12.5% vs. 9.2%, *p* = 0.002) and varied with age. The rate was highest among adults aged >55 years (13.2%), followed by adults aged 18–35 years (11.9%). As for geographical distribution, the highest prevalence of anti-Sars-COV-2 positivity was reported in the administrative departments of Savona (Figure 1). Table 2 shows estimated prevalence according to the three different laboratories.

### 3.3. Factors Associated with Anti-Sars-CoV-2 Antibodies Positivity

Several factors showed an association with anti-SARS-CoV-2 antibodies positivity with univariable analysis (Table 3). The variables that showed a *p*-value < 0.10 were also included in the GEE model (Table 4). The model showed that the main risk factors associated to SARS-CoV-2 seroprevalence were the following: occupational exposure to the virus (OR = 2.36; 95% CI 1.59–3.50, *p* = 0.001), living in a long-term care facility (OR = 4.53; 95% CI 3.19–6.45, *p* = 0.001), and reporting previous symptoms of influenza-like illness (OR = 4.86; 95% CI 3.75–6.30, *p* = 0.001) or loss of sense of smell or taste (OR = 41.00; 95% CI 18.94–88.71, *p* = 0.001).

## 4. Discussion

In the present observational study performed on a large sample of subject in northern Italy, we found the following: (1) the overall seroprevalence of anti-SARS-CoV-2 antibodies (IgG and/or IgM) was 11.0%; (2) occupational exposure to the virus, long-term care facility residency, as well as previous symptoms of influenza-like illness or loss of sense of smell or taste were independently associated with anti-SARS-CoV-2 positivity.

To the best of our knowledge, this is one of the first reports that attempts to describe the prevalence of coronavirus disease and to evaluate the potential circulation of SARS-CoV-2 in North Italy. The findings of our study showed that in a definite geographical area of Italy, approximately 630,000 people might have developed antibodies (11.0% of 5,784,974 inhabitants). This figure is significantly higher than the number of molecular-confirmed SARS-CoV-2 infections (~32,600 cases in the five administrative departments) reported by the Protezione Civile and the Italian National Institute of Health as of 30 April 2020 [2]. The high observed seroprevalence is consistent with recent studies (Table 5) performed in other heavily affected areas of Europe: 9.7% in Geneva, Switzerland [10] and 10.0% in Madrid, Spain [11,12].

Living in a long-term care facility was the strongest predictors of SARS-CoV-2 infection and was reported by 21.6% of anti-SARS-CoV-2-positive participants (*n* = 86/398). This connection was not unexpected [21,22,23], since long-term care facilities often have limited or no infection control programs [24,25] and are usually congregative settings where elderly people have greater exposure to infected patients in the case of respiratory outbreaks [26,27,28]. Therefore, our results emphasized the importance of implementing strategic bundles for infections prevention in long-term care facilities [29]. In this regard, educational interventions on healthcare providers’ knowledge, as well as active surveillance of suspected cases and implementation of barrier precautions, were shown to play a vital role in limiting the spread of other respiratory outbreaks [26,27,28].

Reporting an occupational exposure to the virus also emerged as an independent factor associated with SARS-CoV-2 infection and was reported by 8.7% of anti-SARS-CoV-2-positive participants (*n* = 35/398). However, approximately two-thirds of anti-SARS-CoV-2-positive participants did not report any apparent risk depicting the widespread circulation of the virus in the Italian community, where it has become endemic.

As for clinical symptoms, we found that the prevalence of SARS-CoV-2 antibodies depends on the type of clinical manifestation reported by the patient, being particularly high in people who reported loss of smell or taste [30,31]. Interestingly, 8.6% of participants (*n* = 277/3224) who did not report any symptoms presented antibodies positivity. This finding suggests that non-apparent infection is relatively common in a healthy, active population, thus supporting the hypothesis that, as is true for other coronavirus infections [32], SARS-CoV-2 infection might also be asymptomatic or pauci-symptomatic and resolves spontaneously without any complications in many cases.

In our opinion, the findings of our study could have several implications for pandemic management. Because the real number of patients with SARS-CoV-2 infection is significantly higher than the PCR-confirmed cases, stringent lockdown strategies might possibly be re-implemented only when the intensive care units’ capacities to handle emergencies are overwhelmed. Since a large proportion of patients with SARS-CoV-2 infection are asymptomatic, contract tracing methods to limit the spread of the infection could be particularly challenging. Thus, screening strategies beyond a symptoms-driven approach will be necessary for Italy (e.g., use of mobile applications) to identify enough infected persons to reach SARS-CoV-2 elimination targets [33]; our data could also be useful for vaccine design and implementation.

There are several limitations that should be discussed. Firstly, we do did have any information regarding previous SARS-CoV-2 molecular testing among those patients who tested positive. Accordingly, we cannot provide valuable estimates of antibody prevalence in people positive and negative in PCR testing. Secondly, we analyzed serum samples from patients who voluntarily decided to be tested. Therefore, the clinical characteristics of the sample might differ from those of the general Italian population. Thirdly, geographical prevalence of anti-SARS-CoV-2 antibodies might have been influenced by the type of serological tests used. However, the diagnostic performances of each test are similar to each other; in addition, the highest percentage of infected patients in the Liguria region agrees with recent evidence, suggesting the presence of anti-SARS-CoV-2 antibodies among blood donors from Savona and Genova since December 2019 (unpublished data reported by the Ligurian regional health authority ALISA). Fourthly, all tests we used are non-FDA approved and are yet to be validated. Therefore, prevalence estimates could change once new information on the accuracy of tests are available. Fifthly, the interpretation of the test is still under discussion, because even patients with confirmed SARS-CoV-2 infections have low or non-detectable antibodies titles several weeks after acute infection [34]. Lastly, based on the specificities of testing kits, we cannot exclude that some participants had false positive results due to past or present infection with other viruses, including non-SARS-CoV-2 coronavirus strains [35]. In addition, antibody response may be impaired in elderly, immuno-compromised or immunosuppressed participants, and may produce false negative serology test results [36].

## 5. Conclusions

In conclusion, the results of the present study demonstrate that infection rates based on surveillance data considerably underestimated the infection rates during the SARS-CoV-2 virus pandemic in Italy. The seroprevalence was much higher among people living in long-term care facilities or those with occupational exposure. In our opinion, these findings warrant further investigation into SARS-CoV-2 antibody prevalence among the Italian population.

## Figures and Tables

**Figure 1 jcm-09-02780-f001:**
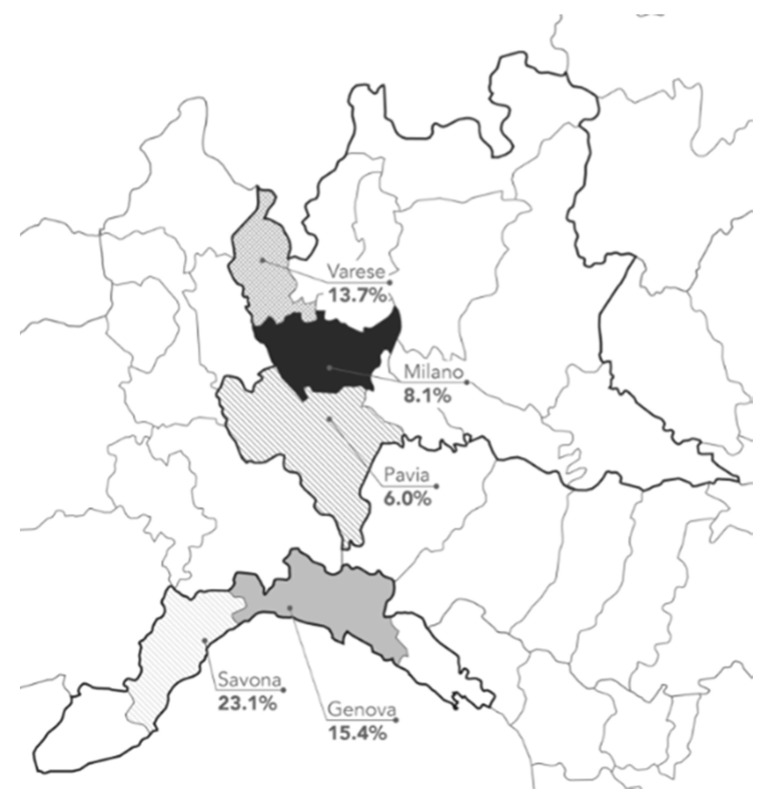
Serologically-confirmed cases of SARS-CoV-2 in the general Italian population from 1 March to 30 April 2020. Red and blue lines represent the boarders of the Lombardia and Liguria regions, respectively. Percentages show the number of positive samples per number tested in each administrative department.

**Table 1 jcm-09-02780-t001:** Clinical characteristics of the study population.

Characteristics	*N* = 3609 (%)
Sex	
Female	2007 (55.6)
Male	1602 (44.4)
Age groups (Years)	
18–35	556 (15.4)
36–45	631 (17.4)
46–55	929 (25.7)
>55	1493 (41.4)
Region	
Lombardia	3065 (84.9)
Liguria	544 (15.1)
Administrative department	
Varese	1430 (39.6)
Pavia	871 (24.1)
Milano	764 (21.2)
Genova	492 (13.6)
Savona	52 (1.4)
Resident in a long-term care facility	207 (5.7)

**Table 2 jcm-09-02780-t002:** Prevalence of SARS-CoV-2 IgM and IgG antibodies according to the three different laboratories.

	*n* (%)	Sars-CoV-2 IgG+ or IgM (95% Confidence Interval)
Medical Center	1885 (52.2)	11.5% (10.1%–13.0%)
Istituto Diagnostico Varelli	1180 (32.7)	8.0% (6.5%–9.7%)
Casa della salute di Genova	544 (15.1)	16.2% (13.2%–19.5%)

**Table 3 jcm-09-02780-t003:** Prevalence of Sars-CoV-2 IgM and IgG antibodies and univariate analysis of factors potentially associated with infection (*n* = 3609).

Characteristics	Tested	SARS-CoV-2 IgG+ or IgM+	Univariate Analysis		
	N	n (%)	OR	95% CI	*p*-Value
Sex					
Female	2007	251 (12.5)	1.36	1.12–1.65	0.002
Male	1602	147 (9.2)	Ref	Ref	Ref
Age group (Years)					
18–35	556	66 (11.9)	1.10	0.83–1.46	0.50
36–45	631	45 (7.1)	0.57	0.41–0.79	0.001
46–55	929	90 (9.7)	0.82	0.64–1.05	0.24
>55	1493	197 (13.2)	1.44	1.17–1.78	0.001
Living in a long-term care facility					
No	3402	312 (9.2)	Ref	Ref	Ref
Yes	207	86 (41.5)	7.56	5.58–10.23	0.001
Occupational exposure					
No	3430	363 (10.6)	Ref	Ref	Ref
Yes	178	35 (19.7)	2.60	1.76–3.88	0.001
Private Exposure					
No	3469	376 (10.8)	Ref	Ref	Ref
Yes	140	21 (15.0)	1.45	0.90–2.36	0.12
Occurrence of Symptoms in the previous month					
No symptoms	3147	226 (7.2)	Ref	Ref	Ref
Influenza-like illness	427	427 (34.2)	6.71	5.27–8.54	0.001
Loss of sense or taste	35	26 (74.3)	37.33	17.28–80.64	0.001

CI Confidence Interval; OR Odd ratio; Ref Reference.

**Table 4 jcm-09-02780-t004:** Results of the generalized estimating equations model of risk factors associated with SARS-CoV-2 seroprevalence.

Characteristics	OR	95% CI	*p*-Value
Male sex	0.79	0.63–1.01	0.06
Age 36–45	1.40	0.99–1.93	0.06
Age > 55	1.17	0.88–1.55	0.27
Living in a long-term care facility	4.53	3.19–6.45	0.001
Occupational exposure	2.36	1.59–3.50	0.001
Prior history of influenza-like illness	4.86	3.75–6.30	0.001
Prior history of loss of sense or taste	41.00	18.94–88.71	0.001

CI Confidence Interval; OR Odd ratio.

**Table 5 jcm-09-02780-t005:** Summary of articles published in the literature reporting data regarding prevalence of SARS-CoV-2 antibodies in the general population.

Author	Country; Area	Number of Participants	Prevalence of Anti-SARS-CoV-2 Antibodies
Petersen M.S. [13]	Faroe Islands; Nationwide study	1075	0.6%
Biggs H. [14]	U.S.; two metropolitan Atlanta counties	696	2.5%
Menachemi N. [15]	U.S; Indiana	3658	2.79%
Fischer B. [16]	Germany; three federal states	3186	0.91%
Pollan M. [11]	Spain; Nationwide study	61,075	5.0%
Havers F. [17]	U.S; 10 sites	16,025	From 1.0% (San Francisco) to 6.9% (New York City)
Amorim Filho L. [18]	Brazil; Rio de Janeiro	2857	4.0%
Percivalle E. [19]	Italy; Lodi area	390	23.0%
Soriano V. [12]	Spain, Madrid	674	13.8%
Stringhini S. [10]	Switzerland, Geneve	2766	9.7%
Sood N. [20]	U.S., Los Angeles	1702	4.3%

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
