# Peer review of "Prevalence of Antibodies to SARS-CoV-2 in Italian Adults and Associated Risk Factors"

_jcm, 2020, doi:10.3390/jcm9092780_

Round 1
Reviewer 1 Report
This is a nice study and informative for the scientific community. There is however a couple of issues that the authors should address before formal acceptance of this paper for publication:
- The authors stated in the discussion that they did not have data on previous PCR-testing. Did the authors ask the participants for previous PCR test for SARS-CoV-2? This could just be an item in the questionnaire. If the authors do not have this information, they could seek it retrospectivelly (I recommend this). If this information becomes available, despite the potential for reporting or recall bias, the authors can provide valuable estimates of antibody prevalence in people positive and negative in PCR testing.
- I think the results should be presented by lab because they used different assays.
- Except lab2, did the other labs use an internal procedure to estimate specificity and sensitivity?
- The specificity of the assay used by lab3 is 90% (and the assay of lab1 has specificity less than 100%), which compromises the results and the prevalence estimates. In general, the authors should apply appropriate statistical methods for adjusting estimates derived from surveys in settings with low disease prevalence and less than optimal specificity.
- The authors should give more details about the three labs (for instance, what kind of tests they routinely perform, how many people they serve on a daily basis, staff numbers etc).
- The authors should apply generalized estimating equations (with logit function) to account for the different panels (labs) in the study.
- Given the fact that the sample is likely not representative of the local population the three labs serve, I think that the authors should avoid extrapolations and strong statements about the actual number of infections and the true infection fatality rate.
- Did the authors get an approval by a bioethics committee?
- The discussion section would improve if the authors performed a comprehensive literature search and presented updated information on estimates of antibody prevalence across different settings.
Author Response
Dear review,
Thank you very much for appreciating our work and for your kind comments. We have carefully made all the modifications suggested by you as follows.
A1. The authors stated in the discussion that they did not have data on previous PCR-testing. Did the authors ask the participants for previous PCR test for SARS-CoV-2? This could just be an item in the questionnaire. If the authors do not have this information, they could seek it retrospectivelly (I recommend this). If this information becomes available, despite the potential for reporting or recall bias, the authors can provide valuable estimates of antibody prevalence in people positive and negative in PCR testing.
As pointed out in our manuscript, we have not collected information on previous PCR testing.
In an attempt to satisfy the reviewer's request, we have tried to collect this information retrospectively. However, patients had to be contacted by phone because a PCR test may also have been performed in a different laboratory than those included in this study.
Despite our effort, it was impossible to re-contact an adequate number of patients in the time available to us for reviewing the manuscript (2 weeks). If the reviewer considers this information as necessary we would be happy to include it but we need to ask to the editor for a longer time (we should contact > 3500 patients by phone). In any case we have further stressed this limitation in our discussion.
A2. I think the results should be presented by lab because they used different assays.
The results have been now presented also by lab as suggested by you. Thank you very much.
A3. Except lab2, did the other labs use an internal procedure to estimate specificity and sensitivity?
All laboratories used internal procedures to estimate the diagnostic performance of serological tests. The results showed values of sensitivity and specificity consistent with those reported by the manufactures. We have now reported this information in our manuscript. Thank you very much!
The specificity of the assay used by lab3 is 90% (and the assay of lab1 has specificity less than 100%), which compromises the results and the prevalence estimates. In general, the authors should apply appropriate statistical methods for adjusting estimates derived from surveys in settings with low disease prevalence and less than optimal specificity.
We are very sorry but we did not understand which statistical method the editor is referring to. Could you be more precise on this point? We would be very happy to apply the suggested statistical method! Thank you again for helping us in improving our manuscript!
The authors should give more details about the three labs (for instance, what kind of tests they routinely perform, how many people they serve on a daily basis, staff numbers etc).
Following your suggestion, we have now included more details about the three labs. Thank you!
A5. The authors should apply generalized estimating equations (with logit function) to account for the different panels (labs) in the study.
Following your suggestion, we have now applied a generalized estimating equation to account for the different panels. The results we have obtained are similar to those previously reported. Thank you very much!
A6. Given the fact that the sample is likely not representative of the local population the three labs serve, I think that the authors should avoid extrapolations and strong statements about the actual number of infections and the true infection fatality rate.
Following your recommendation, we have re-worded some sentences of the discussion avoiding extrapolations or strong statements. Thank you!
A7. Did the authors get an approval by a bioethics committee?
This study has been approved by the ethical committee of the Liguria region. We have now included this important information into the main text reporting the number of ethical approval.
A8. The discussion section would improve if the authors performed a comprehensive literature search and presented updated information on estimates of antibody prevalence across different settings.
We have now included a new table summarizing current literature regarding antibody prevalence in different settings.
We hope to have answered all the queries and comments appropriately. Should you have any further comments or suggestions, please do not hesitate to contact us.
Thank you very much for your help that we fell has improved the quality of our manuscript. We look forward to hearing from you soon.
Sincerely,
Antonio Vena and Matteo Bassetti.
Reviewer 2 Report
This is a very interesting study aiming to assess prevalence of antibodies to SARS-Cov-2. The study is well written and conclusions are supported by the results. Limitations are properly selected. Few comments:
1) It would be interesting to analyse outcomes of patients (minor vs. severe illness or death) and correlate with antibody prevalence.
2) In continuation to the comment above, it is unclear whether patients enrolled in the present study were all healthy and ambulatory or whether there were patients recruited during or after hospital stay.
3) The manuscript needs minor editing for language. Please provide abbreviation meaning before first time an abbreviated term is used in abstract, main text and in figure/table legends.
Author Response
Dear review,
Thank you very much for your kind comments. We have carefully made all the modifications suggested by you as follows:
A1. It would be interesting to analyse outcomes of patients (minor vs. severe illness or death) and correlate with antibody prevalence.
Your consideration is particularly interesting. However, because it is a study addressing the prevalence of antibodies in voluntary participants, none of our patients died because SARS-CoV2. Moreover, we did not collect any information regarding the severity of patients who reported previous history of influenza like illness. We believe that this is an interesting issue that should be addressed in the next future!
A2. In continuation to the comment above, it is unclear whether patients enrolled in the present study were all healthy and ambulatory or whether there were patients recruited during or after hospital stay.
We are very sorry regarding this aspect. We included only voluntary participants in an outpatient setting. We have now tried to more clearly report the inclusion criteria.
A3. The manuscript needs minor editing for language. Please provide abbreviation meaning before first time an abbreviated term is used in abstract, main text and in figure/table legends.
After reviewing the manuscript, we have decided to eliminate all the abbreviation, because they were not strictly necessary. We also carefully revised English language as suggested. Thank you very much!
We hope to have answered all the queries and comments appropriately. Should you have any further comments or suggestions, please do not hesitate to contact us.
Thank you very much for your help that we fell has improved the quality of our manuscript. We look forward to hearing from you soon.
Sincerely,
Antonio Vena and Matteo Bassetti.
Reviewer 3 Report
The authors state that the immunology tests used have not been fully validated. They also state that test interpretations are still under discussion because several PCR+ patients have no detectable antibodies. There should be a fuller discussion of this. How long ago were the PCR tests?
The authors state that serology shows a higher proportion of infected patients than suspected. What is approximate rate of people who got PCR test and what was positive rate? Authors need to reference or support their many assertions.
Author Response
Dear review,
Thank you very much for your kind comments. We have carefully made all the modifications suggested by you as follows:
A1. The authors state that the immunology tests used have not been fully validated. They also state that test interpretations are still under discussion because several PCR+ patients have no detectable antibodies. There should be a fuller discussion of this. How long ago were the PCR tests?
Unfortunately, we have not collected information on previous PCR testing. We have tried to retrospectively collect this data but it was impossible to re-contact an adequate number of patients by phone (we had to contact > 3500 patients). Therefore, we have further stressed this limitation in our discussion. Thank you very much.
A2.The authors state that serology shows a higher proportion of infected patients than suspected. What is approximate rate of people who got PCR test and what was positive rate?Authors need to reference or support their many assertions.
We are very sorry but, as we have not collected data on PCR tests, we are not able to report the positivity rate of patients with suspected infection who have carried out the specific molecular test.
We hope to have answered all the queries and comments appropriately. Should you or the reviewer have any further comments or suggestions, please do not hesitate to contact us.
Thank you very much for your help that we fell has improved the quality of our manuscript. We look forward to hearing from you soon.
Sincerely,
Antonio Vena, Daniele Roberto Giacobbe and Matteo Bassetti.
Reviewer 4 Report
While COVID-19 becomes a world wide pandemic and endemic, it is important to estimate the prevalence in the local population for policy makers and public officers. This study provide preliminary evidence in a regional Italian population. A few comments should be addressed to improve the quality of the study.
- For prevalence study, ideally this should be conducted in public population rather hospital-based population as conducted by this study. Authors should make it clear in the title and throughout the text.
- Section 2.3. Authors should make a table to summarize the performance characteristics of each of the three different assays including the target SARS-CoV-2 proteins (S or N?), antibodies detected, qualitative or quantitative, positive cutoff value etc.
- Section 3.3: Spell out ILI and a foot note for ILI in Table 2.
- While Table 2 summarizes overall results from 5 labs and 3 different assays, author should also show three assay results separately to see the consistency or difference of the prevalence based on different assay performance in the region.
- Authors should discuss the disadvantages of serology tests and the potential impact on prevalence. The following two reference may be cited to strengthen the discussion. Fang B et al. The laboratory's role in combating COVID-19. Crit Rev Clin Lab Sci. 2020 Jun 17:1-15. doi: 10.1080/10408363.2020.1776675. Ghaffari A et al. COVID-19 Serological Tests: How Well Do They Actually Perform? Diagnostics (Basel) 2020 Jul 4;10(7):E453. doi: 10.3390/diagnostics10070453.
Author Response
Dear review,
Thank you very much for your kind comments. We have carefully made all the modifications suggested by you as follows:
A1. For prevalence study, ideally this should be conducted in public population rather hospital-based population as conducted by this study. Authors should make it clear in the title and throughout the text.
We are very sorry if this point was not clear. We included only voluntary participants in an outpatient setting. We have now tried to more clearly report the inclusion criteria.
A2. Section 2.3. Authors should make a table to summarize the performance characteristics of each of the three different assays including the target SARS-CoV-2 proteins (S or N?), antibodies detected, qualitative or quantitative, positive cutoff value etc.
If the reviewer agrees, we would like to avoid adding a new table in our manuscript. As suggested, we have added all the information required in the materials and methods. However, if the editor still believe that an additional table is necessary we will be happy to include it!
A3. Section 3.3: Spell out ILI and a foot note for ILI in Table 2.
We have decided to remove all abbreviations from the text because they were unnecessary. Thank you very much!
A4. While Table 2 summarizes overall results from 5 labs and 3 different assays, author should also show three assay results separately to see the consistency or difference of the prevalence based on different assay performance in the region.
The results have been now presented also by lab as suggested by you. Thank you very much.
Authors should discuss the disadvantages of serology tests and the potential impact on prevalence. The following two reference may be cited to strengthen the discussion. Fang B et al. The laboratory's role in combating COVID-19. Crit Rev Clin Lab Sci. 2020 Jun 17:1-15. doi: 10.1080/10408363.2020.1776675. Ghaffari A et al. COVID-19 Serological Tests: How Well Do They Actually Perform? Diagnostics (Basel) 2020 Jul 4;10(7):E453. doi: 10.3390/diagnostics10070453.
Following your suggestion, we have now discussed the disvantages of serology tests. Thank you very much for your help!
We hope to have answered all the queries and comments appropriately. Should you or the reviewer have any further comments or suggestions, please do not hesitate to contact us.
Thank you very much for your help that we fell has improved the quality of our manuscript. We look forward to hearing from you soon.
Sincerely,
Antonio Vena, Daniele Roberto Giacobbe and Matteo Bassetti.
Round 2
Reviewer 1 Report
No comments